# Itaconate Based Elastomer as a Green Alternative to Styrene–Butadiene Rubber for Engineering Applications: Performance Comparison

**Liwei Li, Haijun Ji, Hui Yang, Liqun Zhang, Xinxin Zhou *** and **Runguo Wang ***

Beijing Advanced Innovation Center for Soft Matter Science and Engineering, State Key Laboratory of Organic-Inorganic Composites, Beijing Laboratory of Biomedical Materials, Beijing University of Chemical Technology, Beijing 100029, China; 2018210268@mail.buct.edu.cn (L.L.); 2020400135@mail.buct.edu.cn (H.J.); 2020400134@mail.buct.edu.cn (H.Y.); zhanglq@mail.buct.edu.cn (L.Z.)

* Correspondence: zhouxinxin2011@126.com (X.Z.); wangrg@mail.buct.edu.cn (R.W.); Tel.: +86-1590-127-3417 (X.Z.); +86-1381-067-5108 (R.W.)

**Abstract:** In response to increasingly stringent requirements for the sustainability and environmental friendliness of the rubber industry, the application and development of bio-based elastomers have received extensive attention. In this work, we prepared a new type of bio-based elastomer poly(dibutyl itaconate-butadiene) copolymer (PDBIB) nanocomposite using carbon black and non-petroleum-based silica with a coupling agent. Using dynamic thermodynamic analysis (DMTA) and scanning electron microscope (SEM), we studied the effects of feed ratio on dynamic mechanical properties, micro morphology, and filler dispersion of PDBIB composites. Among them, silica-reinforced PDBIB60 (weight ratio of dibutyl itaconate to butadiene 40/60) and carbon black-reinforced PDBIB70 (weight ratio of dibutyl itaconate to butadiene 30/70) both showed excellent performance, such as tensile strength higher than 18 MPa and an elongation break higher than 400%. Compared with the widely used ESBR, the results showed that PDBIB had better rolling resistance and heat generation than ESBR. In addition, considering the development of green tires, we compared it with the solution polymerized styrene–butadiene rubber with better comprehensive performance, and analyzed the advantages of PDBIB and the areas to be improved. In summary, PDBIB prepared from bio-based monomers had superior performance and is of great significance for achieving sustainable development, providing a direction for the development of high-performance green tire and holding great potential to replace petroleum-derived elastomers.

**Keywords:** bio-based elastomer; nanocomposites; carbon black; silica; styrene–butadiene rubber

## 1. Introduction

Due to its unique high elasticity and large deformation capacity, rubber materials have become irreplaceable in many fields, such as the tire industry [1–3], sealing industry [4], as well as the damping and shock absorption industry [5–7]. The annual consumption of rubber worldwide exceeds 20 million tons. Thus far, the world has formed a relatively large-scale production of general synthetic rubber [8–10] such as styrene–butadiene rubber, nitrile rubber, neoprene rubber, butadiene rubber, ethylene–propylene rubber, butyl rubber, isoprene rubber, styrene block copolymer thermoplastic elastomers, and special rubber [11,12] represented by polyurethane, fluorine rubber, and silicone rubber. Furthermore, the research and production application systems of all kinds of synthetic rubbers are well developed. At present, synthetic rubber still uses fossil resources such as petroleum as raw materials, which may be sufficient to meet current application needs, but it is not a long-term solution.

Its huge consumption makes the environment and resources greatly threatened, especially in the automotive industry application (tires, hoses, etc.). In future development, it is urgent to develop new high-performance synthetic rubbers that use sustainable resources as raw materials [13,14].

Using renewable resources to prepare a new generation of bio-based elastomer materials is an innovative idea to keep sustainable development of the synthetic rubber industry. At present, two routes are mainly involved. One is to prepare isoprene [15,16], butadiene, and other monomers through biological fermentation processes, and then prepare bio-based isoprene rubbers and other bio-based materials. For example, the Goodyear company has successfully prepared bio-based isoprene rubber, and the LANXESS company has prepared bio-based butyl rubber and ethylene–propylene rubber. The other route is to use bio-based chemicals [13,17] such as propylene glycol, succinic acid, and itaconic acid. Bio-based elastomers are prepared via condensation polymerization [18] or emulsion polymerization [19,20].

Pure synthetic rubber cannot meet engineering applications in terms of mechanical properties and wear resistance, so it needs to be reinforced with nano-fillers. With the development of nanotechnology, a variety of nanoparticles have been developed to be combined with rubber, which not only strengthens the basic mechanical properties of rubber, but also gives the material many functionalities. The widely used nano-fillers mainly include carbon black [21], silica [22], layered silicate [23,24], graphene [25–27], and carbon nanotube [28,29]. Nano-fillers have a significant strengthening effect on rubber materials, which is mainly reflected in the improvement of modulus, tensile strength, and wear resistance. More than 92% of carbon black production in the world was used in rubber manufacturing, especially tire production, such as inner linings, sidewall carcasses, tread, air springs, belts, conveyor wheels, and some vibration isolation devices. At present, the global production of carbon black is about 8.1 million metric tons, ranking among the top 50 industrial chemical manufacturers in the world. However, CB causes pollution because of its origin from petroleum. Silica is a non-petroleum based filler with good prospects. For tires, silica can also reduce the rolling resistance of rubber without sacrificing the wet skid resistance of rubber, that is, it can improve fuel efficiency and reduce greenhouse gas emissions under the premise of ensuring the safe driving of cars. Therefore, silica has great application prospects in "green tire" manufacturing [30]. Due to the surface characteristics of silica, it has a poor affinity with the rubber matrix and is easy to form aggregates, so the key to preparing silica composites is to solve the problem of mixing silica and rubber and dispersion in the rubber matrix. The current methods mainly include in-situ modification technology [31,32], rubber matrix modification [33,34], and wet compounding technology [35]. The continuous development of these technologies has led to the rapid expansion of the silica-filled application. Therefore, the research on the these two types of universal nano-fillers nanocomposites of bio-based elastomers is meaningful for the development of nanofillers and bio-based elastomers.

Bio-based itaconate elastomer (PDBIB) [14,36] involved in this article belongs to the second route. It is a new type of bio-based synthetic rubber prepared based on bulk bio-based chemicals, itaconic acid, n-butanol, and a small amount of diene. At present, the annual output of itaconic acid in China has reached 50,000 tons, and the price is 1662 Dollars/ton. Compared with traditional petrochemical-based chemicals, the cost is no longer a disadvantage. In addition, the static and dynamic mechanical properties of bio-based itaconate elastomers can be flexibly adjusted by adjusting the side groups [37], the copolymerization ratio [36], and the functionalization in the chain [23,38,39]. Moreover, the properties are becoming more perfect. Different from traditional petroleum-based engineering rubber, the molecular structure of bio-based elastomers designed in this paper includes weakly polar long-chain ester groups and flexible butadiene segments, which are quite different from traditional NR (natural rubber), SBR (styrene–butadiene rubber), and NBR (nitrile rubber), so the research about performance and interface interaction of silica-filled PDBIB and carbon black-filled PDBIB are meaningful. Meanwhile, a comparatively comprehensive comparison of the properties of bio-based elastomers nanocomposites and traditional bulk ESBR and SSBR nanocomposites are of great value for the true application of bio-based elastomers.

## 2. Experimental Section

### 2.1. Materials

Dibutyl itaconate (purity of 96%) was purchased from Sigma-Aldrich Company (Burlington, MA, USA). All of the other chemicals and solvents used in the polymerization, including butadiene (purity of 94%), rosin soap, phosphoric acid, potassium hydroxide, ethylene diamine tetraacetic acid (EDTA), sodium dinaphthyllmethanedisulfonate (TAMOL), ferric ethylene diamine tetraacetic acid salt (Fe-EDTA), sodium hydrosulfite (SHS), sodium hydroxymethanesulfinate (SFS), p-menthane hydroperoxide (PMH), hydroxylamine (HA), and ethanol, were kindly provided by Jilin Petroleum Company and used as received. Emlusion-polymerized styrene–butadiene rubber (ESBR 1502) was bought from Tianjin Changli Rubber Trade Co., Ltd. (Tianjin, China) and solution-polymerized styrene–butadiene rubber (SSBR2466) was provided by the TSRC Corporation. The precipitated silica (Ultrasil VN3) with a BET specific surface area of 175 m$^2$/g was bought from Degussa Chemical. All of the rubber additives were industrial grade and commercially available. Deionized water was used for all polymerization runs.

### 2.2. Synthesis of Poly(dibutyl itaconate-co-butadiene) (PDBIB)

As summarized in Table 1, all of the chemicals except SHS and PMH, were added in the sealed reaction bottle, and the mixture in the bottle was pre-emulsified at 25 °C for 4 h to form a stable and homogeneous latex. Later, the SHS solution was injected into the reaction bottle to eliminate the residual oxygen in the pre-emulsion, followed by the PMH solution. The polymerization was allowed to proceed for 8 h at 5 °C, followed by adding hydroxylamine solution to terminate the polymerization, and the PDBIB latex was obtained. The PDBIB gum was obtained by coagulating the latex with ethanol and drying the product in a vacuum oven at 40 °C until a constant weight was obtained.

**Table 1.** Recipe for redox-initiated emulsion polymerization of PDBIB.

| Ingredients (Concentration) | Amount (g) |
| --- | --- |
| Dibutyl itaconate [a] | Various |
| Butadiene [a] | Various |
| Deionized water | 190.00 |
| Disproportionated potassium rosinate | 1.50 |
| Sodium soap | 1.50 |
| Phosphoric acid | 0.23 |
| Potassium hydroxide | 0.40 |
| EDTA | 0.03 |
| TAMOL | 0.14 |
| SHS | 0.04 |
| Fe-EDTA | 0.02 |
| SFS | 0.04 |
| PMH | 0.06 |

[a] The total dosage of the monomers was fixed at 100.00 g, as shown in Table 1.

### 2.3. Preparation of the PDBIB/Silica and ESBR/Silica Nanocomposites

The compounding formulation for PDBIB/silica is shown in Table 2. First, silica with a coupling agent was mixed with neat PDBIB in a HAAKE Rheomix 600 OS internal mixer for 10 min at room temperature. Second, the mixture obtained was rotated in the internal mixer for 5 min at 150 °C to facilitate further reaction between PDBIB and silica, and it was then taken out and cooled down to room temperature. Third, the other additives were mixed with the PDBIB/silica. Finally, the PDBIB/silica compound was cured under 15 MPa at 150 °C for an optimum curing time as determined by a rotor-less rheometer. The ESBR/silica compound was also compounded and vulcanized in the same manner.

**Table 2.** Recipe of PDBIB/silica and ESBR/silica nanocomposites.

| Ingredients. | Loading (phr) [a] |
|---|---|
| PDBIB/ESBR | 70.0 |
| BR9000 | 30.0 |
| Silica (VN3) | 60.0 |
| Si69 | 6.0 |
| Zinc oxide | 5.0 |
| Stearic acid | 2.0 |
| Antioxidant 4020 [b] | 1.0 |
| Antioxidant RD [c] | 1.0 |
| Wax | 1.0 |
| Accelerator CZ [d] | 1.0 |
| Accelerator NS [e] | 1.2 |
| Sulfur | 1.5 |

[a] phr is the abbreviation for weight parts per 100 parts rubber by weight. [b] N-Isopropyl-N′-phenyl-p-phenylene diamine. [c] Poly(1,2-dihydro-2,2,4-trimethyl-quinoline). [d] N-cyclohexylbenzothiazole-2-sulphenamide. [e] N-tert-butylbenzothiazole-2-sulphenamide.

## 2.4. Preparation of the PDBIB/CB and ESBR/CB Nanocomposites

The PDBIB and additives were mixed by a 15.24 cm two-roll mill according to the formulation provided in Table 3. The compound was cured in an XLB-D 350_350 hot press (Huzhou East Machinery, Huzhou, China) under a pressure of 15 MPa at 150 °C for its optimum cure time. ESBR/CB compound was also compounded and vulcanized in the same manner.

**Table 3.** Formulation for the PDBIB/CB and ESBR/CB nanocomposites.

| Ingredient | Loading (phr) |
|---|---|
| PDBIB/ESBR | 100 |
| Carbon black | 60 |
| Zinc Oxide | 3 |
| Stearic acid | 2 |
| TDAE | 10 |
| Antioxidant 4020 | 1 |
| Accelerator DM [a] | 1.2 |
| Accelerator D [b] | 0.6 |
| Sulfur | 1.5 |

[a] 2,2′-dibenzothiazoledisulfde. [b] 1,3-Diphenylguanidine.

## 2.5. Measurements and Characterization

The average molecular weight of PDBIB and ESBR was measured using gel permeation chromatography (GPC) on a Waters Breeze instrument equipped with three water columns (Steerage HT3 HT5 HT6E) using tetrahydrofuran as the solvent (1.0 mL/min) and a Waters 2410 refractive index detector, and polystyrene standards were used for calibration. The FTIR spectra of PDBIB and ESBR were collected on a Tensor 27 spectrometer (Bruker Optic GmbH, Ettlingen, Germany) with the attenuated total reflection (ATR) technique at 4 cm$^{-1}$ resolution with 32 scans under air atmosphere. $^{1}$H NMR measurements of PDBIB and ESBR were conducted with a Bruker AV400 spectrometer (Bruker, Bremen, Germany) with CDCl$_3$ as the solvent. The test frequency was 400 MHz and scanned 16 times. Differential scanning calorimetry (DSC) measurements were performed on a Mettler-Toledo DSC instrument under nitrogen. A sample of 5 mg was heated to 373 K (100 °C) and kept isothermal for 5 min to remove any previous thermal history. Then it was cooled to 173 K (−100 °C) and reheated to 373 K (100 °C) at 10 K/min. The surface morphology of the PDBIB and ESBR nanocomposites were observed in Hitachi S-4800 scanning electron microscope. The samples were prepared by

fracturing the composites in liquid nitrogen and were then sputter coated with gold. The dynamic rheological properties of the PDBIB, ESBR, and SSBR nanocomposites were analyzed using an RPA2000 (Alpha Technologies Co., Hudson, Ohio, USA) at 333 K (60 °C). The strain amplitude was varied from 0.1% to 100% at the test frequency of 1 Hz. The dynamic mechanical properties were determined on a VA 3000 dynamic mechanical thermal analyzer (01 dB-Metravib Co., Rhone Alpes, France) in the tension mode at a fixed frequency of 10 Hz (the most common frequency used to investigate the dynamic mechanical properties in rubber industry, especially in tire industry) and a strain amplitude of 0.3%. The scanned temperature ranged from 193K (−80 °C) to 373 K (100 °C), and the heating rate was of 3 K/min. Tensile tests of the PDBIB, ESBR, and SSBR nanocomposites were conducted according to ASTM D412 (dumbbell-shaped) on a LRX Plus tensile tester made by Lloyd Instruments, Ltd., West Sussex, UK. The interval between vulcanization and testing was 24 h. The test tensile rate was 500 mm/min. The number of samples in the same group was five.

## 3. Results and Discussion

### 3.1. Synthesis and Characterization of PDBIB

A series of bio-based elastomers PDBIB with Bd(butadiene) contents of 50 wt.% to 80 wt.% were prepared using the redox-initiated polymerization carried out under mild conditions. The polymerization equation is illustrated in Scheme 1. The redox-initiated polymerization can obtain high molecular weight polymers as it can decrease the possibility of chain transfer and chain termination. As shown in Table 4, with a variety of Bd(butadiene) content, the number-average molecular weights (Mn) and polydispersity index(PDI) ranged from $33.8 \times 10^4$ g/mol to $45.0 \times 10^4$ g/mol and ranged from 2.97 to 3.79, respectively. We compared it with ESBR and found that the molecular weight of PDBIB was higher than ESBR. The yields of the polymerization were above 65%.

**Scheme 1.** Polymerization reaction of PDBIB.

**Table 4.** Molecular weight and yield of PDBIB and ESBR.

| Sample | $M_n$ ($10^4$) | $M_w$ ($10^4$) | Polydispersity Index | Yield (%) |
|---|---|---|---|---|
| PDBIB50 | 45.0 | 170.5 | 3.79 | 67 |
| PDBIB60 | 40.0 | 133.9 | 3.36 | 70 |
| PDBIB70 | 38.3 | 129.6 | 3.38 | 68 |
| PDBIB80 | 33.8 | 100.2 | 2.97 | 65 |
| ESBR | 11.5 | 44.0 | 3.83 | —— |

### 3.2. Structure Characteristic Comparison of Raw PDBIB and ESBR

Figure 1 displays the FTIR spectra of raw PDBIB polymers with different butadiene contents and ESBR. The absorption peaks at 3017, 2915, and 2840 cm$^{-1}$ were attributed to the stretching vibrations of -CH$_3$, -CH$_2$, and -CH groups in the ESBR, respectively. The peak position of PDBIB here slightly shifted. The absorption at 1640 cm$^{-1}$ belonged to the C=C stretching vibration of the butadiene unit. The difference in the spectra was the absorption at 1728 cm$^{-1}$ and 1175 cm$^{-1}$, which corresponded to the C=O and C-O-C stretching vibration of the DBI (dibutyl itaconate) unit. As the monomer ratio changed (i.e., butadiene gradually increased), we clearly saw that the peak intensities of 1728 cm$^{-1}$ and 1175 cm$^{-1}$ gradually decreased, while the corresponding peak positions of butadiene, such as 1640 cm$^{-1}$ and 969 cm$^{-1}$, gradually increased.

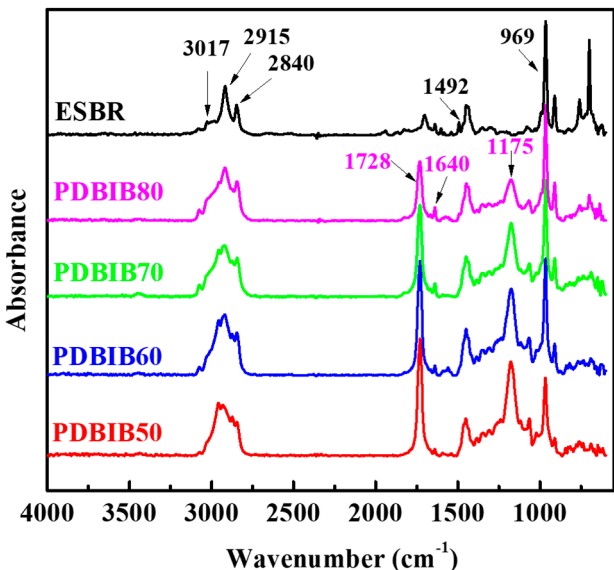

**Figure 1.** FTIR spectra of PDBIB with different butadiene contents and ESBR.

The structural difference of PDBIB and ESBR were further confirmed via $^1$H NMR. The spectra and the detailed assignments of each peak to the molecular structure comparison are shown in Figure 2. The proton shifts of butadiene in their copolymers are similar. The obvious peaks of dibutyl itaconate at 0.94 ppm corresponded to the methyl protons (6) of the side groups and the signals at 4.03 ppm was from the methylene protons (3) of the side groups. Moreover, 7.0–7.3 ppm are attributed to the proton of the benzene ring.

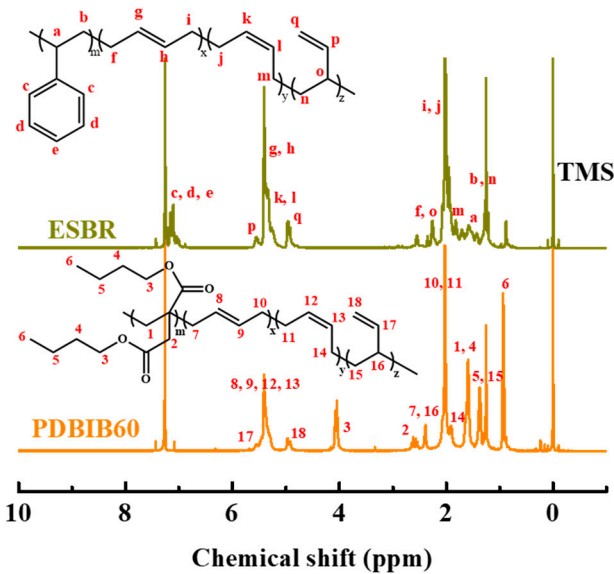

**Figure 2.** $^1$H NMR spectra of PDBIB60 and ESBR.

The elastomeric properties of materials are influenced by the glass transition temperature ($T_g$), which is generally below room temperature for elastomers. Neither of the five curves show any crystallization peak, indicating that the PDBIB and ESBR were all amorphous copolymers, as shown in Figure 3. The $T_g$ values of PDBIB with increased butadiene and ESBR were −57.9, −63.9, −66.2, −70.2, and −53.9 °C, respectively. The reason is that the $T_g$ of styrene homopolymer is much higher than the dibutyl itaconate homopolymer. Simultaneously, it can be found that PDBIB with high butadiene content has better low temperature resistance than ESBR.

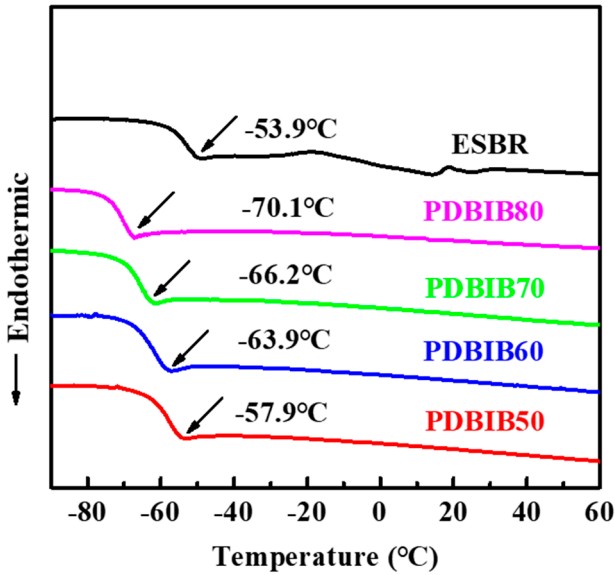

**Figure 3.** DSC thermograms of the synthesized PDBIB and ESBR.

*3.3. Curing Characteristics of PDBIB and ESBR Compounds*

As vulcanization is a critical step for elastomeric products, the effect of filler (silica-VN3 and carbon black-N330) on the curing characteristics was studied. As the modulus of the elastomer increased dramatically during curing, it was used to evaluate the curing process. The curing characteristics of PDBIB and ESBR compounds are shown in Figure 4. As there were many hydroxyl groups on the silica surfaces, polar organic molecules such as accelerator molecules were prone to be adsorbed on them. Therefore, compared with PDBIB/CB or ESBR/CB compounds, PDBIB/silica or ESBR/silica compounds required a longer curing time. In the silica system, PDBIB/silica had a longer scorch time and curing time than ESBR/silica because the polar groups in PDBIB also had an adsorption effect on the accelerator. However, this phenomenon did not appear in the carbon black system.

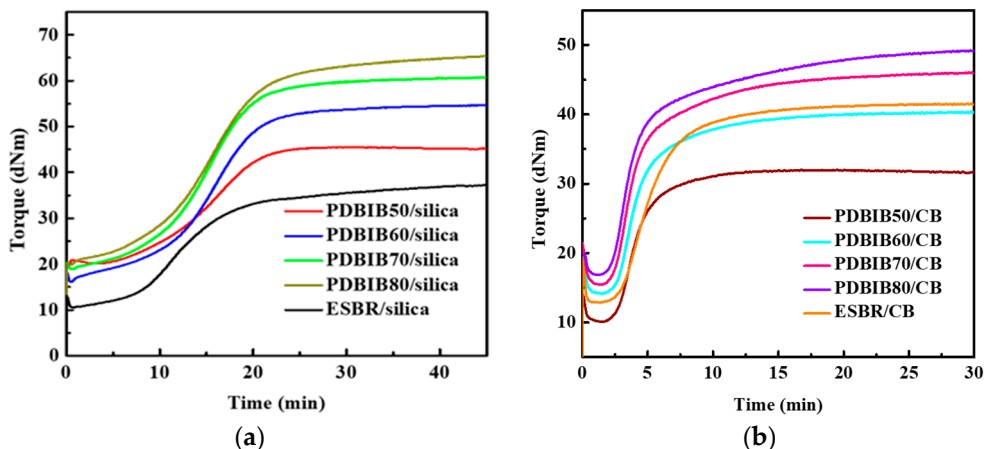

(**a**)　　　　　　　　　　　　　　　　　　　(**b**)

**Figure 4.** Torque of PDBIB and ESBR compounds as a function of curing time (**a**) PDBIB/silica and ESBR/silica; (**b**) PDBIB/CB and ESBR/CB.

The vulcanization characteristics of PDBIB and ESBR compound are shown in Tables 5 and 6. The torque difference became larger, which was caused by the increase in butadiene content. The results indicated the higher degree of crosslinking.

**Table 5.** Curing for the PDBIB/silica and ESBR/silica compound rubber.

| Sample | $t_{10}$/min | $t_{90}$/min | $M_H$-$M_L$ (dN·m) |
|---|---|---|---|
| PDBIB50/silica | 7:03 | 20:39 | 26.3 |
| PDBIB60/silica | 6:27 | 21:36 | 38.6 |
| PDBIB70/silica | 7:20 | 21:02 | 41.8 |
| PDBIB80/silica | 6:43 | 23:35 | 45.7 |
| ESBR/silica | 7:05 | 24:19 | 27.6 |

**Table 6.** Curing for the PDBIB/CB and ESBR/CB compound rubber.

| Sample | $t_{10}$/min | $t_{90}$/min | $M_H$-$M_L$ (dN·m) |
|---|---|---|---|
| PDBIB50/CB | 2:47 | 7:31 | 21.9 |
| PDBIB60/CB | 2:49 | 9:51 | 26.2 |
| PDBIB70/CB | 2:37 | 11:15 | 30.6 |
| PDBIB80/CB | 2:29 | 14:12 | 35.3 |
| ESBR/CB | 3:11 | 9:45 | 28.7 |

### 3.4. Mechanical Properties of PDBIB and ESBR Nanocomposites

Mechanical properties of PDBIB and ESBR nanocomposites are displayed in Figure 5 and summarized in Table 7. In Figure 5a, mechanical properties of PDBIB with various butadiene contents were excellent and comparable to that of ESBR nanocomposites, such as the tensile strength and elongation at break of the PDBIB/silica nanocomposites, all of which exceeded 19 MPa and 450%, respectively. The PDBIB70/silica nanocomposite exhibited the best mechanical properties with a tensile strength of 25.3 MPa and an elongation at break of 536%. The tensile stress at 300% strain and the permanent set are also shown in Table 7. In Figure 5b, since the ester group imparts the polarity of PDBIB, the compatibility with non-polar filler carbon black was not good. The content of ester group was high and carbon was difficult as a reinforcing role, resulting in limited tensile strength. When the introduced butadiene content was higher than 60%, the tensile strength was above 18 MPa, and the elongation at break was above 350%. Compared with ESBR/CB, the tensile strength was not as good as that, but the stress at 300% was obviously higher. In summary, the mechanical properties of PDBIB/silica and PDBIB/CB nanocomposites all could meet most requirements of engineering applications.

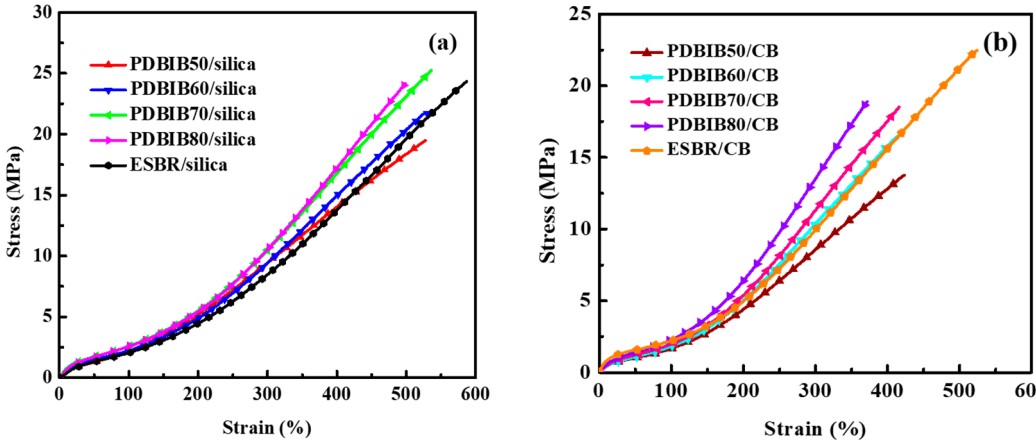

**Figure 5.** Stress–strain curves for PDBIB and ESBR nanocomposites (**a**) PDBIB/silica and ESBR/silica; (**b**) PDBIB/CB and ESBR/CB.

**Table 7.** Mechanical performance of the PDBIB and ESBR nanocomposites.

| Sample | Tensile Strength (MPa) | Stress at 100% (MPa) | Stress at 300% (MPa) | Elongation at Break (%) | Permanent Set (%) | Shore A Hardness |
|---|---|---|---|---|---|---|
| PDBIB50/silica | 19.6 ± 0.4 | 2.2 ± 0.1 | 9.4 ± 0.3 | 527 ± 35 | 12 | 66.1 |
| PDBIB60/silica | 22.0 ± 0.6 | 2.2 ± 0.1 | 9.4 ± 0.4 | 537 ± 23 | 10 | 66.8 |
| PDBIB70/silica | 25.3 ± 0.5 | 2.6 ± 0.1 | 10.5 ± 0.6 | 536 ± 13 | 8 | 68.1 |
| PDBIB80/silica | 24.2 ± 0.4 | 2.6 ± 0.1 | 10.6 ± 0.3 | 498 ± 23 | 6 | 69.5 |
| ESBR/silica | 24.3 ± 0.1 | 2.1 ± 0.1 | 8.5 ± 0.5 | 584 ± 16 | 14 | 65.2 |
| PDBIB50/CB | 13.8 ± 0.8 | 1.7 ± 0.1 | 8.6 ± 0.6 | 424 ± 29 | 16 | 57.2 |
| PDBIB60/CB | 16.5 ± 0.4 | 1.8 ± 0.1 | 10.4 ± 0.3 | 414 ± 25 | 6 | 60.6 |
| PDBIB70/CB | 18.5 ± 0.2 | 2.0 ± 0.1 | 11.4 ± 0.4 | 416 ± 7 | 4 | 60.9 |
| PDBIB80/CB | 18.9 ± 0.8 | 2.3 ± 0.1 | 13.5 ± 0.3 | 372 ± 45 | 2 | 62.1 |
| ESBR/CB | 21.2 ± 0.5 | 2.1 ± 0.1 | 9.4 ± 0.5 | 508 ± 22 | 8 | 64.1 |

*3.5. Filler–Filler and Filler–Polymer Interactions in PDBIB and ESBR Compounds*

In order to analyze the different tensile strength responses of the abovementioned PDBIB/silica and PDBIB/CB with the same filling amount, the filler dispersion and its interaction with PDBIB were analyzed by combining RPA and SEM. As a supplementary note in previous research, it was found that there was a hydrogen bond between itaconate elastomer and silica [19] as Figure 6, which had a positive effect on the dispersion of silica, but there was no such force between ESBR and silica.

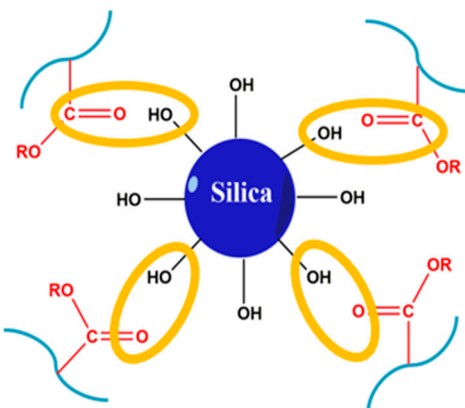

**Figure 6.** Schematic diagram of the hydrogen bonding interaction between itaconate elastomer and silica.

The rubber process analyzer (RPA) allows for the investigation of the strength of a filler–filler network and filler–polymer interactions in a filled elastomeric compound in a wide range of shear strain amplitudes. Figure 7 shows the RPA curves of PDBIB/silica (a) and PDBIB/CB (b) compounds. The storage modulus decreased nonlinearly with increasing applied dynamic strain, which is referred to as the Payne effect [40,41]. As the butadiene moieties of the PDBIB/silica and PDBIB/CB compounds increased, the initial modulus increased, which was caused by the formation of filler–filler interactions. In the Figure 7a, we found that the initial storage modulus of PDBIB was lower than that of ESBR, and the downward trend was gentler. The reasons were as follows: Compared with the benzene ring in the molecular chain structure of ESBR, PDBIB contains polar ester groups. Hydrogen bonding interactions could be formed between silanols on the silica surfaces and ester groups of the macromolecule chains according to our previous studies [42]. The interaction of PDBIB and silica could prevent silica from aggregating to a filler network, resulting in a homogenous dispersion of silica. Meanwhile, the number of ester groups per unit volume decreased with increasing butadiene content in the PDBIB elastomer, which led to the increased initial storage modulus. Therefore, the interaction of PDBIB and silica became weakened, which implied an increased probability for the formation of silica network, resulting in

an increase in the Payne effect and inhomogeneous dispersion of silica. In the Figure 7b, in addition to PDBIB50m which had a more obvious Payne effect, the initial storage modulus and change trend of PDBIB and ESBR were relatively close. Compared with the benzene ring, the ester group has no obvious advantage in the dispersion of carbon black, and even the high content of ester group will affect the dispersion of carbon black. Moreover, the G′ of the PDBIB filled with carbon black was higher than that of them filled with silica.

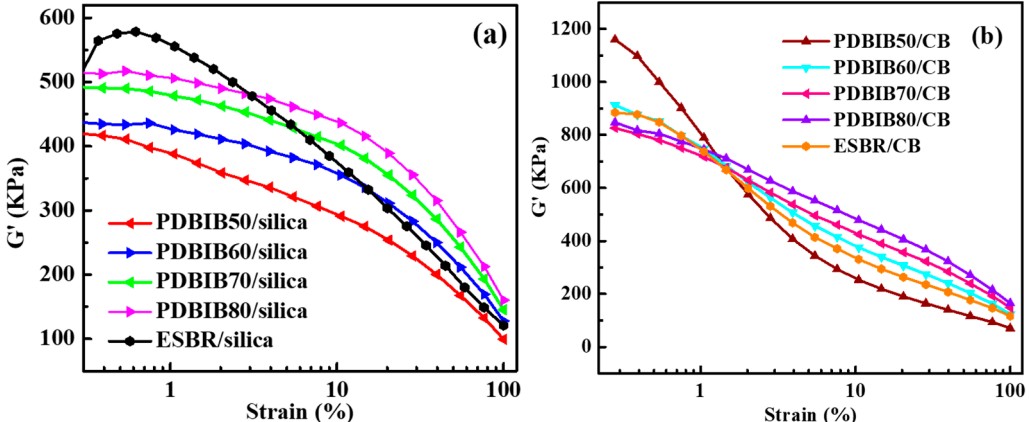

**Figure 7.** Strain dependence of storage modulus (**a**) PDBIB/silica and ESBR/silica, (**b**) PDBIB/CB, and ESBR/CB compounds.

The strain amplitude dependence of tan δ in the PDBIB and ESBR compounds are shown in Figure 8. Destruction of the strong filler–filler network increased filler–filler friction, and weak interfacial interaction increases filler–rubber friction, which resulted in high tan δ at high strain. From Figure 8a, the ester groups of PDBIB helped weaken the filler–filler network and improved the interfacial interaction. Therefore, the tan δ values of PDBIBs/silica were all lower than that of ESBR/silica. The results show that PDBIB elastomers possessed a great potential for future green tire tread applications. In the Figure 8b, it can also be seen that the tan δ values of PDBIBs/CB are lower than that of ESBR/CB, especially PDBIB70 and PDBIB80.

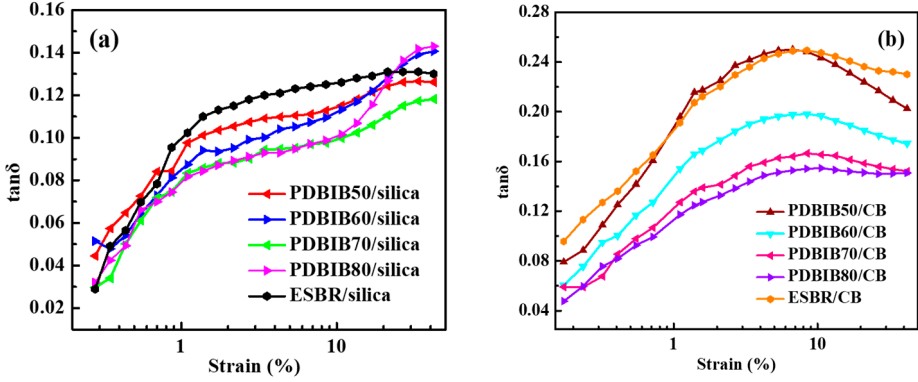

**Figure 8.** Strain dependence of loss tangent (tan δ) (**a**) PDBIB/silica and ESBR/silica, (**b**) PDBIB/CB and ESBR/CB compounds.

### 3.6. Dispersion State of Silica and Carbon Black in the PDBIB and ESBR Nanocomposites

To further compare PDBIB and ESBR nanocomposites, SEM was used to investigate the state of dispersion of fillers (silica or carbon black) in the PDBIB and ESBR nanocomposites. The dark and light parts of the SEM images of Figure 9 represent the PDBIB or ESBR matrix and fillers particles, respectively. The state of dispersion of fillers depended strongly on the filler–polymer interactions. As discussed above, the hydrogen bonding interaction could be formed between silanols and ester

groups of PDBIB60/silica nanocomposites, resulting in more homogenous dispersion of silica compared with ESBR/silica, as shown in Figure 9a,b. Figure 9c,d showed that the dispersion of carbon black in the PDBIB70/CB is more homogenous than that of ESBR/CB.

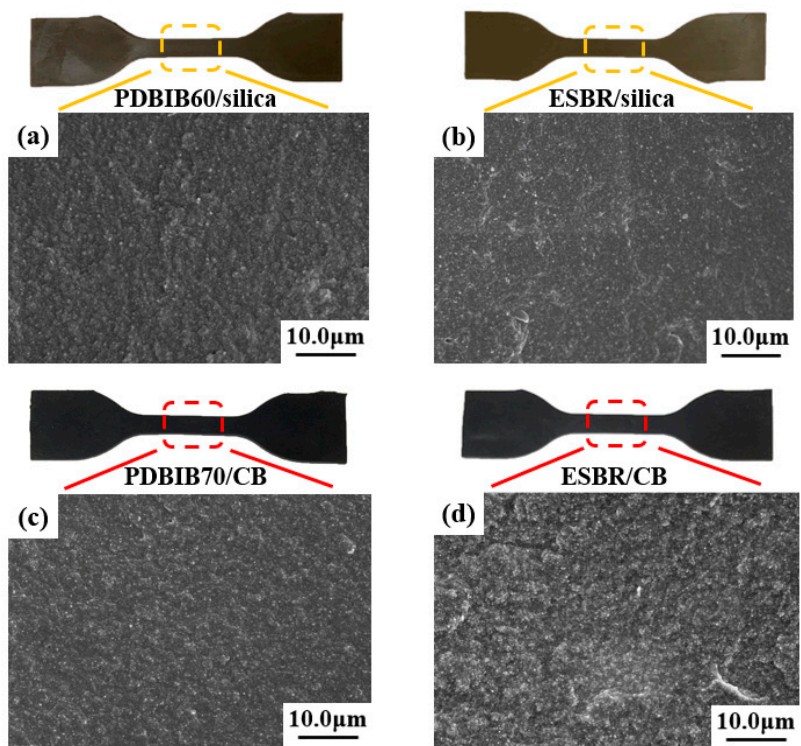

**Figure 9.** SEM micrographs of (**a**) PDBIB60/silica, (**b**) ESBR/silica, (**c**) PDBIB70/CB, and (**d**) ESBR/CB nanocomposites.

### 3.7. Dynamic Mechanical Properties of PDBIB and ESBR Nanocomposites

Dynamic viscoelastic properties were investigated for PDBIB/silica and PDBIB/CB nanocomposites since a majority of engineering elastomers are used under dynamic loading. The value of tan δ at 60 °C was used as a criterion for the rolling resistance, while that at 0 °C was used as a criterion for the wet grip resistance [34]. An ideal material for production of high-performance tires should have a low tan δ value at 60 °C and a high tan δ value at 0 °C. Figure 10 shows the temperature dependence of the tan δ values of the PDBIB/silica and ESBR/silica nanocomposites, and Table 8 gathers most relevant data. From PDBIB50/silica to PDBIB80/silica, the rolling resistance (60 °C tan δ) of PDBIBs/silica decreased, the wet skid resistance (0° C tan δ) also decreased. The tan δ value of PDBIB80/silica nanocomposites was the lowest, indicating that it had excellent low rolling resistance and energy saving characteristics. However, considering the safety, we have taken into account the wet resistance, so we chose PDBIB60/silica, which it had a lower tan δ value at 60 °C and a comparable tan δ values at 0 °C. Compared with ESBR/silica, the dynamic mechanical properties of PDBIB60/silica was more excellent. This is related to the macromolecule structure of PDBIB and ESBR that (1) stronger interfacial interaction between polar ester group and interface; (2) better dispersion of silica.

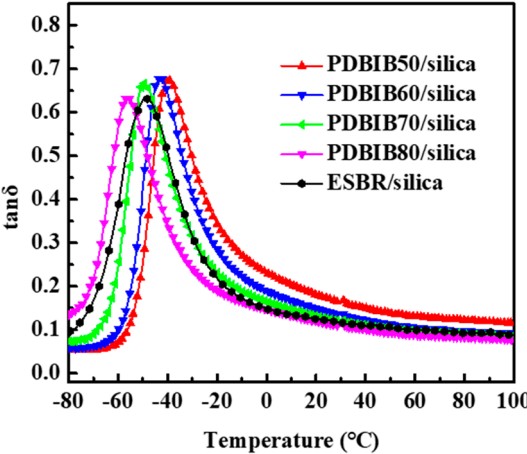

**Figure 10.** Temperature dependence of loss tangent (tan δ) for PDBIB/silica and ESBR /silica nanocomposites.

**Table 8.** Dynamic mechanical thermal analysis of PDBIB/silica and ESBR/silica nanocomposites.

| Sample | $T_g$ (°C) | tan δ | | |
| --- | --- | --- | --- | --- |
| | | 0 °C | 60 °C | Max |
| PDBIB50/silica | −39.2 | 0.230 | 0.131 | 0.680 |
| PDBIB60/silica | −43.3 | 0.189 | 0.105 | 0.685 |
| PDBIB70/silica | −49.3 | 0.165 | 0.097 | 0.669 |
| PDBIB80/silica | −56.4 | 0.146 | 0.088 | 0.637 |
| ESBR/silica | −48.1 | 0.147 | 0.100 | 0.633 |

From Figure 11, the $T_g$ of the PDBIB/CB nanocomposites decreased with increasing butadiene content in the PDBIB, resulting in a decrease in tan δ at both 60 °C and 0 °C for the PDBIB/CB nanocomposite. From Table 9, the tan δ value of PDBIB80/CB nanocomposite was the lowest at 60 °C, indicating the lowest rolling resistance compared to the other PDBIB/CB nanocomposites. Based on the mechanical properties, the wet grip resistance, and the rolling resistance, PDBIB70/CB had the best balanced properties (compared with ESBR/CB, it had a comparable tan δ values at 0 °C and a lower tan δ value at 60 °C). This emerged as a potential elastomer to produce high-performance materials instead of ESBR in tire application.

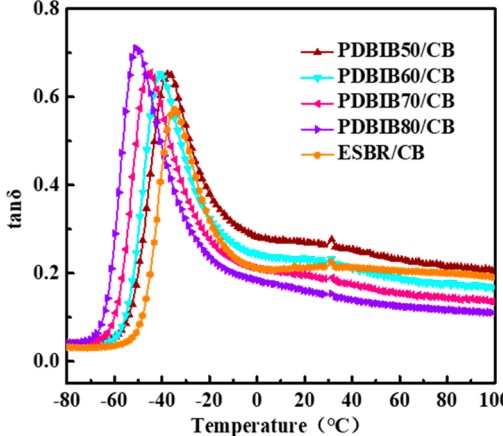

**Figure 11.** Temperature dependence of loss tangent (tan δ) for PDBIB/CB and ESBR/CB nanocomposites.

**Table 9.** Dynamic mechanical thermal analysis of PDBIB/silica nanocomposites.

| Sample | $T_g$ (°C) | tan δ | | |
| --- | --- | --- | --- | --- |
| | | **0 °C** | **60 °C** | **Max** |
| PDBIB50/CB | −36.5 | 0.284 | 0.231 | 0.652 |
| PDBIB60/CB | −41.1 | 0.243 | 0.190 | 0.648 |
| PDBIB70/CB | −45.5 | 0.211 | 0.155 | 0.661 |
| PDBIB80/CB | −50.7 | 0.184 | 0.125 | 0.717 |
| ESBR/CB | −34.4 | 0.211 | 0.206 | 0.571 |

*3.8. Heat Build-Up Test and Abrasion Resistance of PDBIB and ESBR Nanocomposites*

In tire applications, apart from dynamic mechanical properties, the heat build-up test and abrasion resistance are key indicators, and are commonly used to characterize tire performance. In Figure 12a, the heat build-up are arranged from high to low in the following order: ESBR/silica, PDBIB50/silica, PDBIB80/silica, PDBIB60/silica, and PDBIB70/silica. The result aligns with tan δ measured by RPA, except for PDBIB80/silica (Figure 7a). In Figure 12b, we observed that the heat build-up of PDBIB/CB decreased with increasing butadiene content in the PDBIB. Moreover, the heat build-up of all PDBIBs/CB were lower than ESBR/CB.

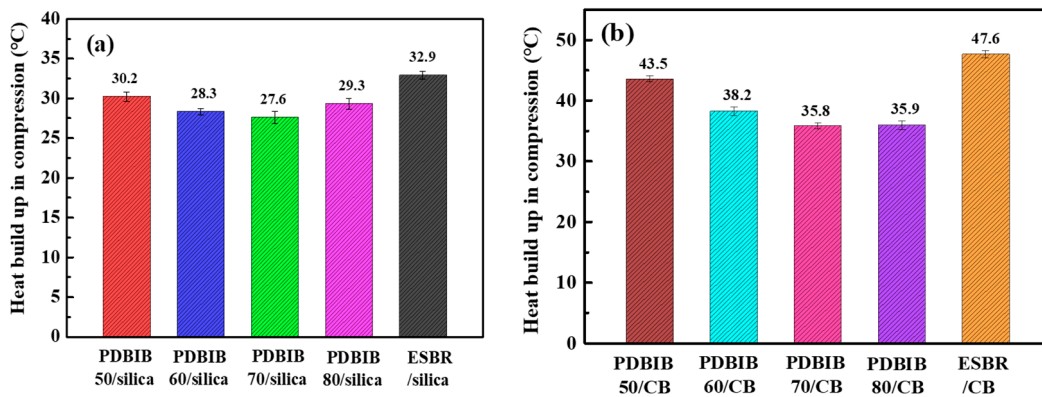

**Figure 12.** Heat build-up (**a**,**b**) of PDBIB and ESBR nanocomposites.

In Figure 13a, as the butadiene moieties increased, the volume loss of abrasion of PDBIB/silica nanocomposites decreased. Among them, abrasion resistance of PDBIB80/silica was excellent and closest to ESBR/silica. In Figure 13b, abrasion resistance of PDBIB70/CB and PDBIB80/CB were better than ESBR/CB. Therefore, bio-based elastomers PDBIB had a great potential for future engineering applications, such as car and truck tires.

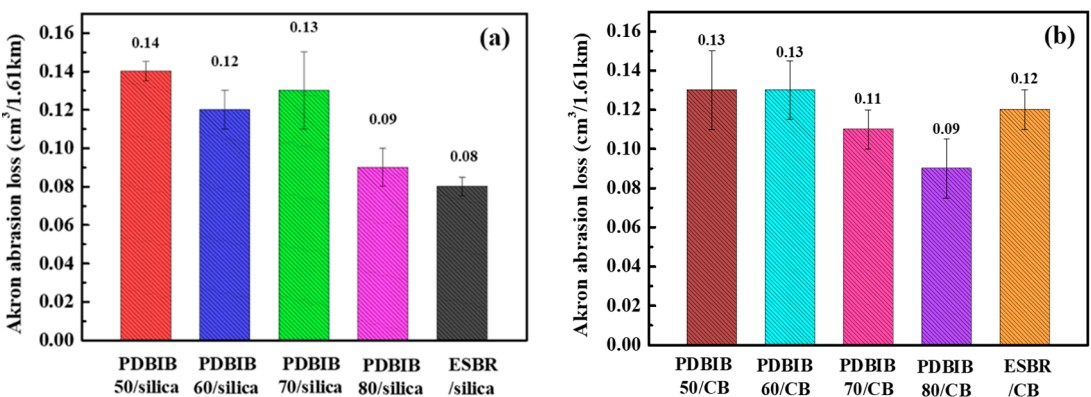

**Figure 13.** Akron abrasion loss (**a**,**b**) of PDBIB and ESBR nanocomposites.

### 3.9. Performance of PDBIB/Silica Nanocomposite Compared with SSBR/Silica Nanocomposite

As mentioned above, PDBIB has a great potential in the tire field, and SSBR is an ideal material for the production of green tires in the modern tire industry. Here, PDBIB60 and SSBR2466 were used to prepare silica-nanocomposites used by the same method described in Section 2.3. Their dynamic and static mechanical properties are shown in Figure 14a,b. The tensile strength and elongation at break of the PDBIB60/silica nanocomposites were higher than the SSBR2466/silica nanocomposites, while its moduli was lower than the SSBR2466/silica nanocomposites. Although the PDBIB60/silica nanocomposites exhibited lower moduli than SSBR2466/silica nanocomposites, they showed mechanical properties that satisfactorily met basic tire requirements. Dynamic properties of PDBIB60/silica and SSBR2466/silica nanocomposites, especially loss tangent (tan δ) values at 60 °C and at 0 °C, were investigated. The tan δ value of the PDBIB60/silica nanocomposite at 60 °C was comparable to the SSBR2466/silica nanocomposite, but it was lower at 0 °C. This is an urgent problem for PDBIB, and we will improve it through functional modification and other means in the future. From Figure 14c,d, we can see that the initial storage modulus of PDBIB60/silica was obviously lower than the SSBR2466/silica, indicating better silica dispersion in PDBIB. Usually, the tan δ value at 7% strain is closely related to the tire rolling resistance [34]. The result showed that the tan δ values at 7% strain in the PDBIB60/silica nanocomposites was a little bit higher than the SSBR2466/silica. Overall, compared with SSBR, the static mechanical properties and filler dispersion of PDBIB are better, but dynamic mechanical properties still need to be improved.

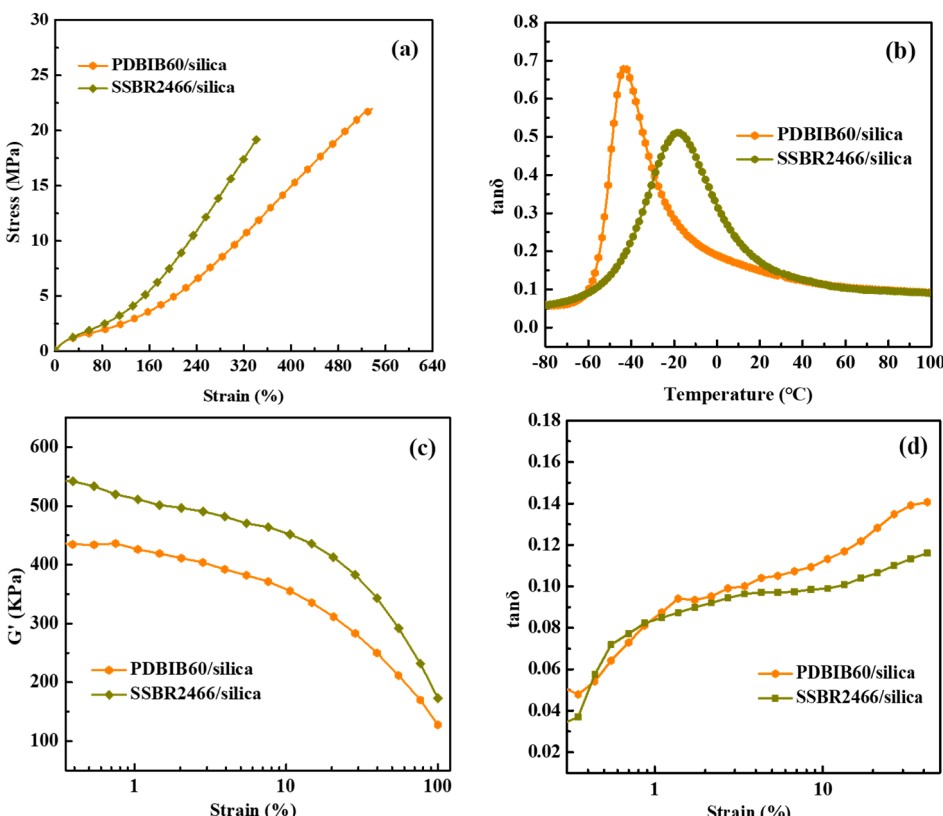

**Figure 14.** Performance comparison of PDBIB and SSBR: (**a**) mechanical properties, (**b**) dynamic mechanical properties, and (**c,d**) strain dependence of storage modulus and loss tangent (tan δ).

## 4. Conclusions

Bio-based cross-linkable poly(dibutyl itaconate-co-butadiene) (PDBIB) elastomers nanocomposites were prepared using silica (VN3) and carbon black (N330). Mechanical properties of PDBIB with different butadiene contents were excellent. The tensile strength and elongation at break of the

PDBIB/silica nanocomposites exceeded 19 MPa and 450%, as did those of the PDBIB70/CB and PDBIB80/CB nanocomposites, exceeding above 18 MPa and 350%. Compared with ESBR, due to the difference of macromolecule structure, polar ester groups in PDBIB could prevent silica from aggregating to a filler network, resulting in a homogenous dispersion of silica. However, the ester group had no obvious advantage in the dispersion of carbon black, and even the high content of ester group could worsen the dispersion of carbon black. In silica-filled and carbon black-filled system, PDBIB60/silica and PDBIB70/CB had the best balanced properties. Moreover, their comprehensive performance were significantly better than the ESBR/filler nanocomposite. Compared with the SSBR, mechanical properties and filler dispersion of PDBIB are better, and dynamic mechanical properties still had space for improvement. On the basis of our results, PDBIBs showed great potential to be a strong and useful supplement for commercial elastomers in engineering applications, especially in the tire industry.

**Author Contributions:** Conceptualization, X.Z., L.Z., and R.W.; formal analysis, L.L. and X.Z.; funding acquisition, R.W.; investigation, L.L., H.J., and H.Y.; project administration, L.Z.; supervision, R.W.; validation, R.W.; writing—original draft, L.L.; writing—review and editing, X.Z. and R.W. All authors have read and agreed to the published version of the manuscript.

**Funding:** The authors would like to thank the National Key Research and Development Program of China (2017YFB0306903), the National Natural Science Foundation of China (51988102, 51503010), and the China–France Cooperation Program of PHC CAI YUANPEI (CHINA SCHOLARSHIP COUNCIL, No. 201504490120) for their financial support.

**Acknowledgments:** The authors would also like to thank Jilin Petrochemical Research Institute for its great help.

**Conflicts of Interest:** The authors declare no conflict of interest.

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
