# Peer review of "Itaconate Based Elastomer as a Green Alternative to Styrene–Butadiene Rubber for Engineering Applications: Performance Comparison"

_processes, doi:10.3390/pr8121527_

Round 1
Reviewer 1 Report
This is a great and complete article. I will only suggested to modify the title which is a bit miss-leading since itaconate is only a comonomer and the elastomer as well most of the properties monitor by the butadiene part
Author Response
Response to Reviewer 1 Comments
Point 1: This is a great and complete article. I will only suggested to modify the title which is a bit miss-leading since itaconate is only a comonomer and the elastomer as well most of the properties monitor by the butadiene part.
Response 1: Thanks so much for the comments. The title “Itaconate Based Elastomer as a Green Alternative to Styrene–butadiene Rubber for Engineering Applications: Performance Comparison” has been used to replace the “Bio-based Itaconate Elastomer as a Green Alternative to Styrene Butadiene Rubber for Engineering Applications: Performance Comparison” in the revised manuscript.
Please see the attachment.

Reviewer 2 Report
Title: Bio-based itaconate elastomer as a green alternative to styrene butadiene rubber for engineering applications: performance comparison
In this work the Authors synthetized dibutyl itaconate/butadiene-based elastomers (PDBIB), using different dibutyl itaconate/butadiene ratios. Those elastomers were also added with two types of fillers (silica and carbon black) during the synthesis, in order to enhance the mechanical properties of the final material. The elastomers were later characterized and compared based on the chosen filler and against a commercial rubber (ESBR).
The aim of the work and the characterization steps are clear and properly described.
However, the presence of several typing errors and paragraphs where the clarity of the description in missing, should be addressed.
Line 45: typing error.
Line 49-61: The Authors should consider rephrasing this paragraph because it lacks clarity.
Line 67: typing error.
Line 89: The Authors should consider adding the price in e.g. Euros or Dollars.
Table 2: there is no reference on the nature of BR9000, which is largely present in the formulation.
Line 189: The Authors should state what abbreviation “Bd” stays for.
Line 191: typing error.
Line 193-194: The Authors should consider rephrasing this sentence because it lacks clarity.
Line 197-198: what stated in this sentence does not match with the results reported in Table 4.
Author Response
Response to Reviewer 2 Comments
Title: Bio-based itaconate elastomer as a green alternative to styrene butadiene rubber for engineering applications: performance comparison
In this work the Authors synthetized dibutyl itaconate/butadiene-based elastomers (PDBIB), using different dibutyl itaconate/butadiene ratios. Those elastomers were also added with two types of fillers (silica and carbon black) during the synthesis, in order to enhance the mechanical properties of the final material. The elastomers were later characterized and compared based on the chosen filler and against a commercial rubber (ESBR).
The aim of the work and the characterization steps are clear and properly described.
However, the presence of several typing errors and paragraphs where the clarity of the description in missing, should be addressed.
Point 1: Line 45: typing error.
Response 1: Thanks so much for the comments. We have made the correction.
Before revision: Its huge consumption makes the environment and resources are greatly threatened, especially in the automotive industry application (tires, hoses, etc.),
After revision: Its huge consumption makes the environment and resources greatly threatened, especially in the automotive industry application (tires, hoses, etc.).
Point 2: Line 49-61: The Authors should consider rephrasing this paragraph because it lacks clarity.
Response 2: Thanks so much for the comments.
Before revision: Using renewable resources to study a new generation of bio-based elastomer materials is an innovative idea to solve the problem of sustainable development of the synthetic rubber industry, and can achieve new breakthroughs in original rubber varieties. At present, two routes are mainly involved. One is to prepare isoprene[15,16], butadiene and other monomers through biological fermentation process, and then prepare bio-based ethylene propylene rubber, bio-based isoprene rubber and other bio-based materials through traditional synthesis process. Traditional rubber has almost the same properties as traditional petroleum-based engineering elastomers and can directly replace existing engineering rubber. Goodyear in the United States has prepared bio-based isoprene rubber, and LANXESS in Germany has also prepared bio-based butyl Rubber and bio-based ethylene propylene rubber; the other is based on bulk bio-based chemicals[13,17] such as propylene glycol, butylene glycol, succinic acid, itaconic acid and other raw materials, prepared by condensation polymerization[18] or emulsion polymerization[19]. It is a new concept synthetic route in the rubber field.
After revision: Using renewable resources to prepare a new generation of bio-based elastomer materials is an innovative idea to keep sustainable development of the synthetic rubber industry. At present, two routes are mainly involved. One is to prepare isoprene[15,16], butadiene and other monomers through biological fermentation process, and then prepare bio-based isoprene rubber and other bio-based materials. For example, Goodyear company has successfully prepared bio-based isoprene rubber, and LANXESS company has prepared bio-based butyl rubber and ethylene propylene rubber. The other route is to use present bio-based chemicals[13,17] such as propylene glycol, succinic acid, itaconic acid. Bio-based elastomers are prepared by condensation polymerization[18] or emulsion polymerization[19].
Point 3:Line 67: typing error.
Response 3: Thanks so much for the comments.
Before revision: The widely used nano-fillers mainly include carbon black[20], silica[21], layered silicate[22,23], graphene[24-26] and carbon nanotube[27,28], etc. Carbon black, this kind of nano-scale particles, has a significant strengthening effect on rubber materials, which is mainly reflected in the modulus, tensile strength, and wear resistance.
After revision: The widely used nano-fillers mainly include carbon black[20], silica[21], layered silicate[22,23], graphene[24-26] and carbon nanotube[27,28]. Nano-fillers have a significant strengthening effect on rubber materials, which is mainly reflected in the improvement of modulus, tensile strength, and wear resistance.
Point 4:Line 89: The Authors should consider adding the price in e.g. Euros or Dollars.
Response 4: Thanks so much for the comments.
Before revision: At present, the annual output of itaconic acid in China has reached 50,000 tons, and the price is 11,000 yuan/ton.
After revision: At present, the annual output of itaconic acid in China has reached 50,000 tons, and the price is 1,662 Dollars/ton.
Point 5:Table 2: there is no reference on the nature of BR9000, which is largely present in the formulation.
Response 5: Thanks so much for the comments. In the article, we used the basic tire formula.
Point 6:Line 189: The Authors should state what abbreviation “Bd” stays for.
Response 6: Thanks so much for the comments. Bd is short for butadiene.
Point 7:line 191: typing error.
Response 7: Thanks so much for the comments.
Before revision: The redox-initiated polymerization can conduce to form high molecular weight polymers due to decrease the possibility of chain transfer and chain termination.
After revision: The redox-initiated polymerization can obtain high molecular weight polymers as it can decrease the possibility of chain transfer and chain termination.
Point 8:Line 193-194: The Authors should consider rephrasing this sentence because it lacks clarity.
Response 8: Thanks so much for the comments. We have made the correction.
Before revision: As shown in Table 4, with variety of Bd content, the number-average molecular weights (Mn) and polydispersity index
(PDI) ranges from 33.8×104 g/mol to 45.0×104 g/mol and ranges from 2.97 to 3.79, respectively.
After revision: As shown in Table 4, with variety of Bd content, the number-average molecular weights (Mn) and polydispersity index (PDI) ranges from 33.8×104 g/mol to 45.0×104 g/mol and ranges from 2.97 to 3.79, respectively.
Point 9:Line 197-198: what stated in this sentence does not match with the results reported in Table 4.
Response 9: Thanks so much for the comments. We have deleted the wrong statement.
Before revision: The yields of the polymerization are above 65%, and the yield decreases with increasing Bd content.
After revision: The yields of the polymerization are above 65%.
Please see the attachment.

Reviewer 3 Report
Dear authors,
thank you for your work! I have some comments on it.
- Generally, you must check the whole manuscript for the use of en dash. For details see section Conflict or connection in https://www.thepunctuationguide.com/en-dash.html
Examples, where en dash must be used are:
styrene–butadiene rubber
ethylene–propylene rubber
filler–filler network
polymer–filler network
stress–strain diagram
Here, no hyphen, but en dash is used.
Furthermore, when you want to express something in the sense of “to”, also en dash must be use. (Example: line 219 7.0–7.3 ppm, see the internet link above).
The theory section is mainly focussed on Chinese publications and this is a form of unilateral literature work. Many groups all over the world are working in the field of e. g. rubber–silica compounds. Therefore, I recommend for the future work to include also these publications to the research to give a complete picture on the state of the art.
- Check the whole manuscript
- for notation of Tg (subscript of g). Not at all places, this is right (examples page 7, line 226 and 227)
- for notation of MPa: p not as capital letter as at page9 in the Table 5.
- if the form of the references to Figures is right; example page 8, line 244 …figure (5-a) instead of “…Figure 5 a”
- for notation of °C; example is at page 14, lines 371/372 where a funny circle was used instead of °.
- for notation of tan; example “Tan” at page 14, Table 7, Table 7
- proper introduction of all abbreviations at the first place where the word to be abbreviated appear; examples: “carbon black” at page 2, line 67, “Bd” which has no introduction, “NR, SBR, NBR” at page , line 96), PDI (Table 4), DBI (line 209)
- for use of SI units; line 144
- for proper use of “rubber” (= raw polymer), rubber mixture/compound (= raw mixture) and elastomer (=vulcanized material). Example: Line 204 è PDBIB elastomers seems to be wrong; I guess, the polymer was investigated by FTIR.
- Remarks about the content
- Lines 70/71: What about the tire tread? This contributes to a high amount to the rolling resistance.
- Line 74: In my opinion, it is too general and wrong, to call silica simply as “green and environmental friendly”.
- Section 2.3/2.4: It is not clear to me, which materials you have prepared. In the Table 2 “BR9000” is mentioned as a component, but there is no explanation. And I miss the composition of the SSBR materials, which are mentioned in 3.8.
- Preparation of the compounds is not clearly described (mixing steps, addition of components, dump time, temperature and torque development). Is the reaction between polymer and silica the silanization reaction? What does ”…was tread for 5 min…” mean?
- Section 2.5 must be revised. From my point of view, no sub headlines are necessary. They disturb the appearance and are completely superfluous. The gained space should be used to inform the reader about more details and results of the methods and measurements. Example: 1H NMR measurements, where no explanation was given about parameters. Second example is tensile test. Here I miss information e. g. about the specimen type, specimen preparation, the test speed, number of specimens per series.
- for notation of tan; example “Tan” at page 14, Table 7
- proper introduction of all abbreviations at the first place where the word to be abbreviated appear; examples: “carbon black” at page 2, line 67, “Bd” (line 189) which has no introduction, “NR, SBR, NBR” at page , line 96), PDI (Table 4), DBI (line 209)
- Section 3.3: Why did you not determine the scorch time and the vulcanization time t90 or t95? It would be easier to discuss the results. Why did you not discuss/mention the different maximum torque levels? What is the reason for this? One aspect may be an influence of the composition on the crosslink density. For this reason, it would be of high importance to characterize the crosslink density. Have you done such investigations to determine the crosslink density of your elastomers? I think, this is an important point, because crosslink density is directly related to the property level.
In Fig. 4, you should change one of the two colors (red or purple) for a better assignability.
- Line 289: It is surprising that the difference in G’ at low and maximum strain amplitude is higher for the CB compounds. Usually, for silica a more pronounced filler networking is observed.
If your goal is to compare silica and CB fillers, I recommend the same axis for the two series.
- Section 3.6: from my point of view it is critical to characterize the filler dispersion only from a fracture surface by SEM. Usually, this is done by light microscopy (macro dispersion) or by TEM (micro dispersion). This means, the presented results cannot be discussed in a general manner. Furthermore, the filler dispersion does not depend only on the polymer–filler interaction, but e. g. also on the amount of filler and the mixing conditions.
- The sections 3.5 and 3.7 contain results from DMA tests. My question is: why it is divided into different sections?
I would recommend a revision of the structure of section 3 and a new order of the results in this section. You have section 3.4, where you present mechanical test results, and then in section 3.7 abrasion loss is presented. This is not fitting. I could imagine 3 sub sections: “vulcanization behaviour and filler dispersion” “physical and mechanical properties” “viscoelastic properties”. However, this depends on the aim of the investigations! You should formulate this clearly, and then you can arrange the results in an adequate structure and order (as an example: you may think about another arrangement: Fig. 11 a together with Fig. 13 a and Fig. 11 b together with Fig. 13b.).
In this sense, also the section 3.8 is questionable. There was no information about the materials' composition and therefore no discussion about the differences in ESBR and SSBR. The reader must ask: Why you present suddenly results from tests with SSBR materials? The goal must be clearly named and described.
- Mistakes / linguistic formulation
- line 57, rubber instead if Rubber
- line 123: obtained instead of obtain
- line 182: K instead of °C
- Sentences on line 191/192 is unclear.
- line 83/84: What does “bulk nano-fillers composite mean?
- Line 172: It sounds that the sputtering was done before fracturing.
- Line 194/195 unnecessary break
- End of line 196: “…higher than that of ESBR.”
- Lines 210 to 212: space characters are missing between the number and the unit.
- Line 219: “7.0–7.3 ppm are” or “… the peak …is…”
- In the head of Table 5 something is shifted-
- Line 280: “Meantime…” I do not understand the sentences.
- Line 283: Payne effect without hyphen.
- Lines 295/296: In the figure are no ester groups. Therefore, the sentences must be for example like this: “From Fig.
Best regards!
Author Response
Response to Reviewer 3 Comments
Dear authors,
thank you for your work! I have some comments on it.
Point 1:Generally, you must check the whole manuscript for the use of en dash. For details see section Conflict or connection in https://www.thepunctuationguide.com/en-dash.html.
Examples, where en dash must be used are:
styrene–butadiene rubber
ethylene–propylene rubber
filler–filler network
polymer–filler network
stress–strain diagram
Here, no hyphen, but en dash is used.
Furthermore, when you want to express something in the sense of “to”, also en dash must be use. (Example: line 219 7.0–7.3 ppm, see the internet link above).
The theory section is mainly focussed on Chinese publications and this is a form of unilateral literature work. Many groups all over the world are working in the field of e. g. rubber–silica compounds. Therefore, I recommend for the future work to include also these publications to the research to give a complete picture on the state of the art.
Response 1: Thanks so much for the comments. We have checked the entire article and have revised the wrong expression.
Before revision: ethylene propylene rubber, styrene-butadiene rubber, filler-filler, filler-polymer, Stress-strain
After revision: ethylene–propylene rubber, styrene–butadiene rubber, filler–filler, filler–polymer, stress–strain
Before revision: The peaks at 7.0-7.3 ppm is attributed to the proton of the benzene ring.
After revision: 7.0–7.3 ppm are attributed to the proton of the benzene ring.
Point 2:Check the whole manuscript for notation of Tg (subscript of g). Not at all places, this is right (examples page 7, line 226 and 227)
Response 2: Thanks so much for the comments. We have checked the entire article and have revised the wrong expression.
The elastomeric properties of materials are influenced by the glass transition temperature (Tg), which is generally below room temperature for elastomers. Neither of the five curves show any crystallization peak, indicating that the PDBIB and ESBR were all amorphous copolymers, as shown in Figure 3. The Tg values of PDBIB with increased butadiene and ESBR were -57.9, -63.9, -66.2, -70.2 and -53.9 °C, respectively. The reason is that the Tg of styrene homopolymer is much higher than that of dibutyl itaconate homopolymer. Simultaneously, it can be found that PDBIB with high butadiene content has better low temperature resistance than ESBR.
Point 3:for notation of MPa: p not as capital letter as at page9 in the Table 5.
Response 3: Thanks so much for the comments. We have revised the wrong expression.
|
Sample |
Tensile strength /MPa |
Stress at 100%/ MPa |
Stress at 300%/ MPa |
Elongation at break /% |
Permanent set /% |
Shore A hardness |
Point 4:if the form of the references to Figures is right; example page 8, line 244 …figure (5-a) instead of “…Figure 5 a”
Response 4: Thanks so much for the comments. We have checked the entire article and have revised the wrong expression. We have modified “In Figure (5-a)” to “In Figure 5 a”.
Point 5:for notation of °C; example is at page 14, lines 371/372 where a funny circle was used instead of °.
Response 5: Thanks so much for the comments. We have checked the entire article and have made the correction.
Point 6:for notation of tan; example “Tan” at page 14, Table 7, Table 7
Response 6: Thanks so much for the comments. We have modified “Tan” to “tan”.
Point 7:proper introduction of all abbreviations at the first place where the word to be abbreviated appear; examples: “carbon black” at page 2, line 67, “Bd” which has no introduction, “NR, SBR, NBR” at page , line 96), PDI (Table 4), DBI (line 209)
Response 7: Thanks so much for the comments. We have checked the entire article and have made the correction. We have used “Bd (butadiene)” instead of “Bd”, and “NR (natural rubber), SBR (styrene–butadiene rubber), NBR (nitrile rubber)” instead of “NR, SBR and NBR”, “polydispersity index” instead of “PDI”, and “DBI (dibutyl itaconate)” instead of “DBI”.
Point 8:for use of SI units; line 144
Response 8: Thanks so much for the comments.
Before revision: The PDBIB and additives were mixed by a 6-in two-roll mill according to the formulation provided in Table 3.
After revision: The PDBIB and additives were mixed by a 15.24 cm two-roll mill according to the formulation provided in Table 3.
Point 9:for proper use of “rubber” (= raw polymer), rubber mixture/compound (= raw mixture) and elastomer (=vulcanized material). Example: Line 204 è PDBIB elastomers seems to be wrong; I guess, the polymer was investigated by FTIR.
Response 9: Thanks so much for the comments.
Before revision: Figure 1 displays the FTIR spectra of raw PDBIB elastomers with different butadiene contents and ESBR.
After revision: Figure 1 displays the FTIR spectra of raw PDBIB polymers with different butadiene contents and ESBR.
Remarks about the content
Point 10:Lines 70/71: What about the tire tread? This contributes to a high amount to the rolling resistance.
Response 10: Thanks so much for the comments.
Before revision: More than 92% of carbon black production in the world was used in rubber manufacturing, especially tire production, such as inner linings, sidewall carcasses, air springs, belts, conveyor wheels and some vibration isolation devices.
After revision: More than 92% of carbon black production in the world was used in rubber manufacturing, especially tire production, such as inner linings, sidewall carcasses, tread, air springs, belts, conveyor wheels and some vibration isolation devices.
Point 11:Line 74: In my opinion, it is too general and wrong, to call silica simply as “green and environmental friendly”.
Response 11: Thanks so much for the comments.
Before revision: Silica is a green and environmentally friendly rubber-reinforced filler with good prospects.
After revision: Silica is a non-petroleum based filler with good prospects.
Point 12:Section 2.3/2.4: It is not clear to me, which materials you have prepared. In the Table 2 “BR9000” is mentioned as a component, but there is no explanation.
Response 12: Thanks so much for the comments. In the article, we used the basic tire formula.
Point 13:And I miss the composition of the SSBR materials, which are mentioned in 3.8. Preparation of the compounds is not clearly described (mixing steps, addition of components, dump time, temperature and torque development). Is the reaction between polymer and silica the silanization reaction? What does ”…was tread for 5 min…” mean?
Response 13: Thanks so much for the comments.
Preparation of the compounds is the same as 2.3. Here, PDBIB60 and SSBR2466 were used to prepare silica-nanocomposites used by the same method as 2.3.
The reaction between polymer and silica is the silanization reaction.
The mixture obtained was rotated in the internal mixer for 5 min at 150°C to facilitate further reaction between PDBIB and silica, and it was then taken out and cooled down to room temperature.
Point 14: Section 2.5 must be revised. From my point of view, no sub headlines are necessary. They disturb the appearance and are completely superfluous. The gained space should be used to inform the reader about more details and results of the methods and measurements. Example: 1H NMR measurements, where no explanation was given about parameters. Second example is tensile test. Here I miss information e. g. about the specimen type, specimen preparation, the test speed, number of specimens per series.
Response 14: Thanks so much for the comments.
Before revision:1H NMR measurements of PDBIB and ESBR were conducted with a Bruker AV400 spectrometer (Bruker, Germany) with CDCl3 as the solvent.
After revision: 1H NMR measurements of PDBIB and ESBR were conducted with a Bruker AV400 spectrometer (Bruker, Germany) with CDCl3 as the solvent. The test frequency is 400 MHz, scanning 16 times.
Before revision: Tensile tests of the PDBIB、ESBR and SSBR nanocomposites were conducted according to ASTM D412 (dumbbell-shaped) on a LRX Plus tensile tester made by Lloyd Instruments, Ltd., UK. The interval between vulcanization and testing was 24 h.
After revision: Tensile tests of the PDBIB、ESBR and SSBR nanocomposites were conducted according to ASTM D412 (dumbbell-shaped) on a LRX Plus tensile tester made by Lloyd Instruments, Ltd., UK. The interval between vulcanization and testing was 24 h. The test tensile rate is 500 mm/min. The number of samples in the same group is five.
Point 15:for notation of tan; example “Tan” at page 14, Table 7
Response 15: Thanks so much for the comments. We have checked the entire article and have revised the wrong expression.
Point 16: proper introduction of all abbreviations at the first place where the word to be abbreviated appear; examples: “carbon black” at page 2, line 67, “Bd” (line 189) which has no introduction, “NR, SBR, NBR” at page , line 96), PDI (Table 4), DBI (line 209)
Response 16: Thanks so much for the comments. We have checked the entire article and have made the correction. We have used “Bd (butadiene)” instead of “Bd”, and “NR (natural rubber), SBR (styrene–butadiene rubber), NBR (nitrile rubber)” instead of “NR, SBR and NBR”, “polydispersity index” instead of “PDI”, and “DBI (dibutyl itaconate)” instead of “DBI”.
Point 17:Section 3.3: Why did you not determine the scorch time and the vulcanization time t90 or t95? It would be easier to discuss the results. Why did you not discuss/mention the different maximum torque levels? What is the reason for this? One aspect may be an influence of the composition on the crosslink density. For this reason, it would be of high importance to characterize the crosslink density. Have you done such investigations to determine the crosslink density of your elastomers? I think, this is an important point, because crosslink density is directly related to the property level.
Response 17: Thanks so much for the comments. We have discussed the scorch time, vulcanization time and torque. The torque difference can reflect the crosslink density. And we have also made the following changes in the article.
The vulcanization characteristics of PDBIB and ESBR compound are shown in Table 5 and Table 6. The torque difference becomes larger and larger, which is caused by the increase in butadiene content. The result indicated the higher degree of crosslinking.
Table 5. Curing for the PDBIB/silica and ESBR/silica compound rubber
|
Sample |
T10/min |
T90/min |
MH-ML(dN·m) |
|
PDBIB50/silica |
7:03 |
20:39 |
26.3 |
|
PDBIB60/silica |
6:27 |
21:36 |
38.6 |
|
PDBIB70/silica |
7:20 |
21:02 |
41.8 |
|
PDBIB80/silica |
6:43 |
23:35 |
45.7 |
|
ESBR/silica |
7:05 |
24:19 |
27.6 |
Table 6. Curing for the PDBIB/CB and ESBR/CB compound rubber
|
Sample |
T10/min |
T90/min |
MH-ML(dN·m) |
|
PDBIB50/CB |
2:47 |
7:31 |
21.9 |
|
PDBIB60/CB |
2:49 |
9:51 |
26.2 |
|
PDBIB70/CB |
2:37 |
11:15 |
30.6 |
|
PDBIB80/CB |
2:29 |
14:12 |
35.3 |
|
ESBR/CB |
3:11 |
9:45 |
28.7 |
Point 18:In Fig. 4, you should change one of the two colors (red or purple) for a better assignability.
Response 18: Thanks so much for the comments. We have revised the misleading figure.
Point 19:Line 289: It is surprising that the difference in G’ at low and maximum strain amplitude is higher for the CB compounds. Usually, for silica a more pronounced filler networking is observed.
If your goal is to compare silica and CB fillers, I recommend the same axis for the two series.
Response 19: Thanks so much for the comments. Because this is a new type of elastomer. Different from traditional petroleum-based engineering rubber, the molecular structure of bio-based elastomers includes polar long-chain ester groups and flexible butadiene segments, which are quite different from traditional NR, SBR and NBR. Therefore, CB compared with Silica, a more pronounced filler networking is observed.
In this article, our main purpose is to compare the performance of PDBIB and ESBR, not two fillers.
Point 20:Section 3.6: from my point of view it is critical to characterize the filler dispersion only from a fracture surface by SEM. Usually, this is done by light microscopy (macro dispersion) or by TEM (micro dispersion). This means, the presented results cannot be discussed in a general manner. Furthermore, the filler dispersion does not depend only on the polymer–filler interaction, but e. g. also on the amount of filler and the mixing conditions.
Response 20: Thanks so much for the comments. We use RPA and SEM to study the dispersion state of fillers in this article. SEM is a partly analysis method to characterize the filler dispersion. The storage modulus decrease content in RPA results show good relationship with the filler dispersion. We discussed this part in Figure 7 and Figure 8.
Point 21:The sections 3.5 and 3.7 contain results from DMA tests. My question is: why it is divided into different sections?
Response 21: Thanks so much for the comments. Because we use two fillers, they belong to two systems.
Point 22:I would recommend a revision of the structure of section 3 and a new order of the results in this section. You have section 3.4, where you present mechanical test results, and then in section 3.7 abrasion loss is presented. This is not fitting. I could imagine 3 sub sections: “vulcanization behaviour and filler dispersion” “physical and mechanical properties” “viscoelastic properties”. However, this depends on the aim of the investigations! You should formulate this clearly, and then you can arrange the results in an adequate structure and order (as an example: you may think about another arrangement: Fig. 11 a together with Fig. 13 a and Fig. 11 b together with Fig. 13b.).
Response 22: Thanks so much for the comments. We have revised the part.
3.7. Dynamic mechanical properties of PDBIB and ESBR nanocomposites
Dynamic viscoelastic properties were investigated for PDBIB/silica and PDBIB/CB nanocomposites since a majority of engineering elastomers are used under dynamic loading. The value of tan δ at 60°C is used as a criterion for the rolling resistance, while that at 0°C is used as a criterion for the wet grip resistance[33]. An ideal material for production of high-performance tires should have a low tan δ value at 60°C and a high tan δ value at 0°C. Figure 10 shows the temperature dependence of the tan δ values of the PDBIB/silica and ESBR/silica nanocomposites, and Table 8 gathers most relevant data. From PDBIB50/silica to PDBIB80/silica, the rolling resistance (60°C tan δ) of PDBIBs/silica decreases, the wet skid resistance (0°C tan δ) also decreases. The tan δ value of PDBIB80/silica nanocomposites is the lowest which indicated it had the excellent low rolling resistance and energy saving characteristics. But considering the safety, we have taken into account the wet resistance, so we chose PDBIB60/silica, which it had a lower tan δ value at 60 °C and a comparable tan δ values at 0 °C. Compared with ESBR/silica, dynamic mechanical properties of PDBIB60/silica was more excellent. This is related to the macromolecule structure of PDBIB and ESBR that (1) stronger interfacial interaction between polar ester group and interface; (2) better dispersion of silica.
Figure 10. Temperature dependence of loss tangent (tan δ) for PDBIB/silica and ESBR/silica nanocomposites.
Table 8. Dynamic mechanical thermal analysis of PDBIB/silica and ESBR/silica nanocomposites.
|
sample |
Tg (oC) |
tan δ |
||
|
0oC |
60oC |
Max |
||
|
PDBIB50/silica |
-39.2 |
0.230 |
0.131 |
0.680 |
|
PDBIB60/silica |
-43.3 |
0.189 |
0.105 |
0.685 |
|
PDBIB70/silica |
-49.3 |
0.165 |
0.097 |
0.669 |
|
PDBIB80/silica |
-56.4 |
0.146 |
0.088 |
0.637 |
|
ESBR/silica |
-48.1 |
0.147 |
0.100 |
0.633 |
The Tg of the PDBIB/CB nanocomposites decreased with increasing butadiene content in the PDBIB, resulting in a decrease in tan δ at both 60°C and 0°C for the PDBIB/CB nanocomposite. From Table 9, the tan δ value of PDBIB80/CB nanocomposite was the lowest at 60°C, indicating the lowest rolling resistance compared to the other PDBIB/CB nanocomposites. Based on the mechanical properties, the wet grip resistance, and the rolling resistance, PDBIB70/CB had best balanced properties (compared with ESBR/CB, it had a comparable tan δ values at 0 °C and a lower tan δ value at 60 °C.) and this emerged as a potential elastomer to produce high-performance materials instead of ESBR in tire application.
Figure 11. Temperature dependence of loss tangent (tan δ) for PDBIB/CB and ESBR/CB nanocomposites.
Table 9. Dynamic mechanical thermal analysis of PDBIB/silica nanocomposites.
|
sample |
Tg (oC) |
tan δ |
||
|
0oC |
60oC |
Max |
||
|
PDBIB50/CB |
-36.5 |
0.284 |
0.231 |
0.652 |
|
PDBIB60/CB |
-41.1 |
0.243 |
0.190 |
0.648 |
|
PDBIB70/CB |
-45.5 |
0.211 |
0.155 |
0.661 |
|
PDBIB80/CB |
-50.7 |
0.184 |
0.125 |
0.717 |
|
ESBR/CB |
-34.4 |
0.211 |
0.206 |
0.571 |
3.8. Heat build-up test and abrasion resistance of PDBIB and ESBR nanocomposites
In tire application, apart from dynamic mechanical properties, the heat build-up test and abrasion resistance are also very key indicators, which used to characterize tire performance. In Figure 12 a, the heat build-up are arranged from high to low in the following order: ESBR/silica, PDBIB50/silica, PDBIB80/silica, PDBIB60/silica and PDBIB70/silica. The result aligns with tan δ measured by RPA except for PDBIB80/silica (Figure 7 a). In Figure 12 b, we can see that the heat build-up of PDBIB/CB decreased with increasing butadiene content in the PDBIB. Moreover, the heat build-up of all PDBIBs/CB were lower than that of ESBR/CB.
|
|
|
|
Figure 12. Heat build-up (a) and (b) of PDBIB and ESBR nanocomposites
|
|
In Figure 13 a, as the butadiene moieties increased, the volume loss of abrasion of PDBIB/silica nanocomposites decreases. Among them, abrasion resistance of PDBIB80/silica was excellent and was closest to that of ESBR/silica. In Figure 13 b, abrasion resistance of PDBIB70/CB and PDBIB80/CB were better than that of ESBR/CB. Therefore, biobased elastomers PDBIB has a great potential for future engineering applications, such as car, truck tires.
|
Figure 13. Akron abrasion loss (a) and (b) of PDBIB and ESBR nanocomposites. |
|
Point 23:In this sense, also the section 3.8 is questionable. There was no information about the materials' composition and therefore no discussion about the differences in ESBR and SSBR. The reader must ask: Why you present suddenly results from tests with SSBR materials? The goal must be clearly named and described.
Response 23: Thanks so much for the comments. The information about the materials' composition is in the Materials.
Because the current low rolling resistance green tires mainly use SSBR, we compare SSBR to explore the application potential of PDBIB in green tires.
Mistakes / linguistic formulation
Point 24:line 57, rubber instead if Rubber
Response 24: Thanks so much for the comments. We have modified “Rubber” to “rubber”.
Point 25:line 123: obtained instead of obtain
Response 25: Thanks so much for the comments. We have modified “obtained” to “obtain”.
Point 26:line 182: K instead of °C
Response 26: Thanks so much for the comments.
Before revision: The scanned temperature ranged from -80 to 100℃, and the heating rate was of 3℃/min.
After revision: The scanned temperature ranged from 193K (-80℃) to 373K (100℃), and the heating rate was of 3K/min.
Point 27:Sentences on line 191/192 is unclear.
Response 27: Thanks so much for the comments.
Before revision: The redox-initiated polymerization can conduce to form high molecular weight polymers due to decrease the possibility of chain transfer and chain termination.
After revision: The redox-initiated polymerization can obtain high molecular weight polymers as it can decrease the possibility of chain transfer and chain termination.
Point 28:line 83/84: What does “bulk nano-fillers composite mean?
Response 28: Thanks so much for the comments.
Before revision: Therefore, the research on the these two types of bulk nano-fillers nanocomposites of bio-based elastomers is meaningful for the development of nanofillers and bio-based elastomers.
After revision: Therefore, the research on the these two types of universal nano-fillers nanocomposites of bio-based elastomers is meaningful for the development of nanofillers and bio-based elastomers.
Point 29:Line 172: It sounds that the sputtering was done before fracturing.
Response 29: Thanks so much for the comments.
Before revision: The samples were prepared by fracturing the composites in liquid nitrogen and were previously sputter coated with gold.
After revision: The samples were prepared by fracturing the composites in liquid nitrogen and were then sputter coated with gold.
Point 30:Line 194/195 unnecessary break
Response 30: Thanks so much for the comments. We have revised the wrong expression.
As shown in Table 4, with variety of Bd content, the number-average molecular weights (Mn) and polydispersity index(PDI) ranges from 33.8×104 g/mol to 45.0×104 g/mol and ranges from 2.97 to 3.79, respectively.
Point 31:End of line 196: “…higher than that of ESBR.”
Response 31: Thanks so much for the comments.
Before revision: We compared it with ESBR and found that the molecular weight of PDBIB is much higher than ESBR.
After revision: We compared it with ESBR and found that the molecular weight of PDBIB is much higher than that of ESBR.
Point 32:Lines 210 to 212: space characters are missing between the number and the unit.
Response 32: Thanks so much for the comments. We have revised the wrong expression.
The absorption at 1640 cm-1 belongs to the C=C stretching vibration of the butadiene unit. The difference in the spectra is the absorption at 1728 cm-1 and 1175 cm-1, which corresponds to the C=O and C-O-C stretching vibration of the DBI unit. As the monomer ratio changes (butadiene gradually increases), we can clearly see that the peak intensities of 1728 cm-1 and 1175 cm-1 gradually decrease, while the corresponding peak positions of butadiene, such as 1640 cm-1 and 969 cm-1, gradually increase.
Point 33:Line 219: “7.0–7.3 ppm are” or “… the peak …is…”
Response 33: Thanks so much for the comments. We have revised the wrong expression.
Before revision: The peaks at 7.0-7.3 ppm is attributed to the proton of the benzene ring.
After revision: 7.0–7.3 ppm are attributed to the proton of the benzene ring.
Point 34:In the head of Table 5 something is shifted-
Response 34: Thanks so much for the comments. We have revised the wrong table.
Table 5. Mechanical performance of the PDBIB and ESBR nanocomposites.
|
Sample |
Tensile strength /MPa |
Stress at 100%/ MPa |
Stress at 300%/ MPa |
Elongation at break /% |
Permanent set /% |
Shore A hardness |
|||
|
PDBIB50/silica |
19.6±0.4 |
2.2±0.1 |
9.4±0.3 |
527±35 |
12 |
66.1 |
|
||
|
PDBIB60/silica |
22.0±0.6 |
2.2±0.1 |
9.4±0.4 |
537±23 |
10 |
66.8 |
|
||
|
PDBIB70/silica |
25.3±0.5 |
2.6±0.1 |
10.5±0.6 |
536±13 |
8 |
68.1 |
|
||
|
PDBIB80/silica |
24.2±0.4 |
2.6±0.1 |
10.6±0.3 |
498±23 |
6 |
69.5 |
|
||
|
ESBR/silica |
24.3±0.1 |
2.1±0.1 |
8.5±0.5 |
584±16 |
14 |
65.2 |
|
||
|
PDBIB50/CB |
13.8±0.8 |
1.7±0.1 |
8.6±0.6 |
424±29 |
16 |
57.2 |
|
||
|
PDBIB60/CB |
16.5±0.4 |
1.8±0.1 |
10.4±0.3 |
414±25 |
6 |
60.6 |
|
||
|
PDBIB70/CB |
18.5±0.2 |
2.0±0.1 |
11.4±0.4 |
416±7 |
4 |
60.9 |
|
||
|
PDBIB80/CB |
18.9±0.8 |
2.3±0.1 |
13.5±0.3 |
372±45 |
2 |
62.1 |
|
||
|
ESBR/CB |
21.2±0.5 |
2.1±0.1 |
9.4±0.5 |
508±22 |
8 |
64.1 |
|
||
Point 35:Line 280: “Meantime…” I do not understand the sentences.
Response 35: Thanks so much for the comments.
Before revision: Meantime, also observed the number of ester groups per unit volume decreased with increasing butadiene content in the PDBIB elastomer.
After revision: Meantime, the number of ester groups per unit volume decreased with increasing butadiene content in the PDBIB elastomer, which leaded to the increased initial storage modulus.
Point 36:Line 283: Payne effect without hyphen.
Response 36: Thanks so much for the comments. We have modified “Payne effect” to “Payne‑effect”.
Point 37:Lines 295/296: In the figure are no ester groups. Therefore, the sentences must be for example like this: “From Fig.
Response 37: Thanks so much for the comments.
Before revision: In the figure (8-a), the ester groups of PDBIB help to weaken the filler-filler network and improve the interfacial interaction.
After revision: From Figure 8 a, the ester groups of PDBIB help to weaken the filler-filler network and improve the interfacial interaction.
Please see the attachment.

Reviewer 4 Report
a well written, relevant paper.
Author Response
Thanks for your comments.
Round 2
Reviewer 3 Report
Hello,
thank you for t he revision.
I have some further (minor) comments:
Line 173: Write 10 K/min instead of 10°C/min.
Line 251 and 253: Symbol of time is t not T (being the temperature). Furthermore, 10 and 90 must be subscript.
Line 259: In the head of the table, the last letter of elongation and permanent is at the next line.
Line 294: Check "leaded". I guess, it is led.
To point 13 of the first author's report: Thank you for the explanation for me, but I did not found new text/explanation in the manuscript.
To point 14: Unfortunately, the subheadlines in 2.5 are still there.
Best regards!
Author Response
Response to Reviewer 3 Comments
Dear Reviewer:
Thank you for your comments.
Point 1:Line 173: Write 10 K/min instead of 10°C/min.
Response 1: Thanks so much for the comments. We have made the correction.
Point 2:Line 251 and 253: Symbol of time is t not T (being the temperature). Furthermore, 10 and 90 must be subscript.
Response 2: Thanks so much for the comments. We have modified “T10”and “T90” to “t10” and “t90”.
Point 3:Line 259: In the head of the table, the last letter of elongation and permanent is at the next line.
Response 3: Thanks so much for the comments. We have modified the table.
|
Sample |
Tensile strength /MPa |
Stress at 100%/ MPa |
Stress at 300%/ MPa |
Elongation at break /% |
Permanent set /% |
Shore A hardness |
Point 4:Line 294: Check "leaded". I guess, it is led.
Response 4: Thanks so much for the comments. We have modified “leaded” to “led”.
Point 5:To point 13 of the first author's report: Thank you for the explanation for me, but I did not found new text/explanation in the manuscript.
Response 5: Thanks so much for the comments. We have made the correction.
Before revision: Here, PDBIB60 and SSBR2466 were used to prepare silica-nanocomposites.
After revision: Here, PDBIB60 and SSBR2466 were used to prepare silica-nanocomposites used by the same method as 2.3.
Point 6:To point 14: Unfortunately, the subheadlines in 2.5 are still there.
Response 6: Thanks so much for the comments. We have made the correction.
2.5. Measurements and characterization
The average molecular weight of PDBIB and ESBR was measured by gel permeation chromatography (GPC) on a Waters Breeze instrument equipped with three water columns (Steerage HT3 HT5 HT6E) using tetrahydrofuran as the solvent (1.0 mL/min) and a Waters 2410 refractive index detector, and polystyrene standards were used for calibration. The FTIR spectra of PDBIB and ESBR were collected on a Tensor 27 spectrometer (Bruker Optic GmbH, Germany) with the attenuated total reflection (ATR) technique at 4 cm-1 resolution with 32 scans under air atmosphere.1H NMR measurements of PDBIB and ESBR were conducted with a Bruker AV400 spectrometer (Bruker, Germany) with CDCl3 as the solvent. The test frequency is 400 MHz, scanning 16 times. Differential scanning calorimetry (DSC) measurements were performed on a Mettler-Toledo DSC instrument under nitrogen. A sample of 5 mg was heated to 373 K(100°C) and kept isothermal for 5 min to remove any previous thermal history. Then it was cooled to 173 K(-100°C) and reheated to 373 K(100°C) at 10 K/min. The surface morphology of the PDBIB and ESBR nanocomposites were observed in Hitachi S-4800 scanning electron microscope. The samples were prepared by fracturing the composites in liquid nitrogen and were then sputter coated with gold. The dynamic rheological properties of the PDBIB、ESBR and SSBR nanocomposites were analyzed by an RPA2000 (Alpha Technologies Co., USA) at 333 K(60℃). The strain amplitude was varied from 0.1% to 100% at the test frequency of 1 Hz. The dynamic mechanical properties were determined on a VA 3000 dynamic mechanical thermal analyzer (01 dB-Metravib Co., France) in the tension mode at a fixed frequency of 10 Hz (the most common frequency used to investigate the dynamic mechanical properties in rubber industry, especially in tire industry) and a strain amplitude of 0.3%. The scanned temperature ranged from 193K (-80℃) to 373K (100℃), and the heating rate was of 3K/min. Tensile tests of the PDBIB、ESBR and SSBR nanocomposites were conducted according to ASTM D412 (dumbbell-shaped) on a LRX Plus tensile tester made by Lloyd Instruments, Ltd., UK. The interval between vulcanization and testing was 24 h. The test tensile rate is 500 mm/min. The number of samples in the same group is five.
Please see the attachment.